# Dramatic Deterioration of Subclinical Hyperparathyroidism in Children and Adolescents During the Post-COVID-19 Period

**DOI:** 10.3390/diseases13070198

**Published:** 2025-06-27

**Authors:** Maria Loutsou, Eleni Dermitzaki, Rodis D. Paparodis, Aspasia N. Michoula, Nicholas Angelopoulos, Panagiotis Christopoulos, Stavros Diamantopoulos, George Mastorakos, Ioanna N. Grivea, Dimitrios T. Papadimitriou

**Affiliations:** 1Department of Pediatrics, School of Medicine, University of Thessaly, General University Hospital of Larissa, Biopolis, 41110 Larissa, Greece; amichoula@gmail.com (A.N.M.); ioanna.grivea@gmail.com (I.N.G.); 2Pediatric Endocrine Clinics, Athens Pediatric Center, Marousi, 15125 Athens, Greece; dermitzaki.elen@gmail.com; 3Division of Endocrinology, Diabetes and Metabolism, Loyola University Medical Center, Maywood, IL 60153, USA; rodis@paparodis.gr; 4Hellenic Endocrine Network, Ermou 6 Str., 10563 Athens, Greece; drangelnick@gmail.com; 5Second Department of Obstetrics and Gynecology, Medical School, “Aretaieion” University Hospital, National and Kapodistrian University of Athens, 11527 Athens, Greece; dr_christopoulos@yahoo.gr; 6Pediatric Endocrine Division, Iaso Children’s Hospital, 11521 Athens, Greece; sdiamantop@gmail.com; 7Areteio Hospital, Medical School, National and Capodistrian University of Athens, 10674 Athens, Greece; gmastorak@med.uoa.gr

**Keywords:** vitamin D, vitamin D deficiency, vitamin D insufficiency, subclinical hyperparathyroidism, normocalcemic hyperparathyroidism, bone health, rickets

## Abstract

**Background:** Vitamin D is a steroid hormone, essential for the immune system and bone health. Since the sun is meant to provide at least 80% of daily vitamin D requirements, the COVID-19 pandemic is likely to have induced a considerable influence on calcium metabolism. **Methods:** We analyzed data from 1138 children, seen in an outpatient pediatric endocrinology clinic from 2022–2023. Vitamin D status was classified as deficiency if 25(OH)D ≤ 20 ng/mL, insufficiency < 30 ng/mL, and sufficiency ≥ 30 ng/mL. **Results:** Overall, 60.8% of children had vitamin D deficiency or insufficiency worsened with age (*p* < 0.005), and with adolescent males having higher 25(OH)D concentrations than females (*p* < 0.05). A negative correlation was found between 25(OH)D and BMI SDS (*R*^2^ = 0.02, *p* < 0.001), and 25(OH)D concentrations varied seasonally, decreasing in winter. Subclinical hyperparathyroidism [parathyroid hormone (PTH) > 45 pg/mL) and normal calcium] was found in 21.5% of children, with 73.5% of them being vitamin D deficient or insufficient. A negative correlation between PTH and 25(OH)D was observed, with PTH plateauing at 25(OH)D above 40 ng/mL (*p* < 0.001). **Conclusions:** Compared to the pre-pandemic data (2016–2018), with only 5.1% of children having subclinical hyperparathyroidism (*p* < 0.001), these findings suggest a marked deterioration in vitamin D status and calcium metabolism in children, with possible unforeseen consequences for bone, immune, and general health.

## 1. Introduction

Vitamin D, also known as the sunshine vitamin, is the only vitamin produced through sunlight exposure, making dietary intake of minor importance provided sufficient sun exposure. As a supplement, it exists in two forms: ergocalciferol (D2) and cholecalciferol (D3), both lipophilic, but with modern cholecalciferol preparations being water-soluble [1].

Vitamin D is essential for bone health and calcium absorption. It promotes the intestinal absorption of calcium and phosphorus while reducing their renal excretion. These effects are mainly mediated through the VDR in the cell nucleus, acting as a transcription factor to regulate gene expression involved in calcium homeostasis and immune system regulation [2]. The VDR is found in almost all tissues, including the heart, brain, skin, and breast (Figure 1). Vitamin D aids in bone mineralization by interacting with PTH and regulates bone resorption via the RANK ligand pathway [2]. Conditions like hypocalcemia or vitamin D deficiency lead to increased PTH (hyperparathyroidism) [3]. Vitamin D deficiency has been linked to an increased risk of infection, as well as hypertension, metabolic syndrome, cancer and autoimmune diseases, such as type 1 diabetes, multiple sclerosis, and even rheumatoid arthritis and inflammatory bowel disease [2]. During the first wave of the COVID-19 pandemic, higher vitamin D concentrations were associated with a reduced incidence of severe disease and lower mortality [4]. Vitamin D sufficiency maintains bone health and prevents rickets [5], and probably higher concentrations than 30 ng/mL are required to achieve extraskeletal actions [6,7].

Vitamin D insufficiency (<30 ng/mL) and deficiency (<20 ng/mL) can range from asymptomatic to severe symptomatic rickets with secondary hyperparathyroidism and osteoporosis [10]. Vitamin D deficiency is now considered a pandemic, affecting over 1 billion people worldwide [11,12]. In a Greek study in 2019, involving 3773 adults, the mean 25(OH)D was 16.67 ng/mL and 16.74 ng/mL in men and women, respectively [12]. In another study involving 55,844 Europeans, 40.4% had 25(OH) < 20 ng/mL [13]. Another study conducted in Greece showed vitamin D deficiency at 57.7% with apparent predominance in the winter months [14], up to 92.2% in the pre-COVID era (January–March 2017) [15]. Despite a marked increase in 25(OH)D laboratory testing and supplementation in recent years—particularly after 2020—serum 25(OH)D concentrations have remained persistently low. Real-world data indicate that even long-term supplementation (>12 months) with daily doses of 3000 IU results in vitamin D sufficiency in only 67.7% of individuals [16], consistent with “The big Vitamin D mistake” [17]. A study conducted among elderly individuals in Greece identified 32 ng/mL as the threshold concentration of 25-hydroxyvitamin D [25(OH)D], below which parathyroid hormone (PTH) levels begin to rise. Approximately 20% of the elderly participants were found to have vitamin D deficiency accompanied by elevated PTH levels, indicating secondary hyperparathyroidism [18]. Furthermore, a cohort study conducted between 2007 and 2009 involving 2386 Greek children aged 9–13 years revealed that 52.5% of participants were deficient in 25(OH)D. Notably, the deficiency was more pronounced during the spring than in the winter months [19].

Several vitamins have been investigated for their potential role in supporting immune function and reducing the risk or severity of COVID-19. Among them, vitamin D has the strongest clinical backing, with studies suggesting that deficiency is associated with increased susceptibility to infection, greater symptom severity, and worse outcomes; supplementation may offer benefits, though results are mixed [4]. Vitamin C, known for its antioxidant properties, has shown promise in some trials—particularly in intravenous form—for reducing inflammation and improving oxygenation in severe cases, though evidence remains inconclusive [20]. Vitamin A may support mucosal immunity and lung health, while vitamin E acts as an antioxidant and may be beneficial in older populations, though direct evidence in COVID-19 is limited. B vitamins (especially B6, B9, and B12) are thought to support cellular immunity and reduce inflammation, but clinical data on their role in COVID-19 remain sparse [20]. In another meta-analysis, it was suggested that vitamin D supplementation may help reduce COVID-19-related mortality and ICU admissions, though it does not significantly lower the risk of initial infection [20]. Vitamin C, particularly in high intravenous doses, showed potential benefits in improving oxygenation and reducing inflammation in severe cases, though the evidence remains inconclusive. Vitamins A, B, and E may support immune function, but direct clinical evidence of their efficacy against COVID-19 is limited [21]. Overall, while certain vitamins—especially vitamin D—appear to offer some therapeutic value in managing disease severity, more thorough research is required to confirm these effects and establish clear clinical guidelines.

In the last 15 years, a new entity in the spectrum of hyperparathyroidism, subclinical hyperparathyroidism, has been recognized. It is a condition characterized by elevated PTH > 45 ng/mL, normal blood calcium levels in three consecutive measurements over a 3–6 month period, and exclusion of other causes of secondary hyperparathyroidism [22]. A significant number of studies have been published since 2008 aiming to further characterize this disorder and its incidence [23,24]. In a study with an 8-year follow-up of patients with a primary diagnosis of subclinical hyperparathyroidism, 75% were later shown to have secondary hyperparathyroidism [23], suggesting that subclinical hyperparathyroidism is the precursor of the classic form of hyperparathyroidism, as, in approximately 15% of patients, it can progress to tertiary hyperparathyroidism. In 1988, Rao et al. first proposed a biphasic progression of hyperparathyroidism [25]. They suggested that PTH levels are elevated during the initial phase, but serum calcium remains within the reference range. This first phase is generally asymptomatic but was not recognized as a precursor or early form of hyperparathyroidism until recently. The second phase involves hypercalcemia, along with elevated PTH levels, and is considered classic primary hyperparathyroidism [25]. In Greece, before the COVID-19 pandemic (2016–2018), between 3060 children tested, 5.1% of them were found with subclinical hyperparathyroidism (PTH > 45 pg/mL), with 40% of them being vitamin D deficient. In most of these patients, vitamin D and, where needed, calcium supplementation normalized completely PTH [26].

The aim of the present study was to assess key parameters of calcium metabolism—including the incidence of vitamin D deficiency, insufficiency, and subclinical hyperparathyroidism—in children and adolescents during the post-COVID-19 period (2022–2023) compared to corresponding data from the pre-COVID era (2016–2018).

## 2. Materials and Methods

Data were collected fully anonymously from children examined for the first time at the pediatric endocrinology outpatient clinic in 2022 and 2023, using the Growth Analyzer EPRS software (ver 4.2.7). Sex, age, month of examination, Tanner pubertal staging, BMI SDS (Body Mass Index Standard Deviation Score), and serum concentrations of Ca (calcium), P (phosphorus), ALP (alkaline phosphatase), 25(OH)D, and PTH (parathyroid hormone) were retrieved where available. Classification into preadolescence and adolescence was according to the Tanner criteria (B2 for girls and G2 for boys, using a Pradder orchidometer (>3 mL) in boys [27]. Vitamin D status was based on the Endocrine Society guidelines [28], as deficiency [25(OH)D ≤ 20 ng/mL], insufficiency [25(OH)D < 30 ng/mL], and sufficiency [25(OH)D ≥ 30 ng/mL]. Intact PTH levels were measured by second-generation immunoassays [29], and thresholds were defined as, low ≤ 15 pg/mL, optimal PTH ≤ 35 pg/mL, normal PTH < 45 pg/mL, elevated PTH ≥ 45 pg/mL, and high ≥ 65 pg/mL [30]. Patients with a history of disease affecting Ca and vitamin D metabolism (liver, renal and metabolic bone diseases, inflammatory bowel disease, celiac disease, and malabsorption syndromes), as well as those on vitamin D supplements for any reason, were excluded from the study. Statistical analysis was performed with the XLSTAT PREMIUM-AI ver. 2023.1.4 (copyright Addinsoft).

From 2032 patients’ records, 1138 children meeting the above criteria had available calcium metabolism parameters (986 all of them) and were enrolled in the analysis. Their mean (± SD) age was 9.3 ± 3.75 years [range: 1 month–16 years]. Specifically, 631/1138 (55%) were boys and 507/1138 (45%) were girls, with 269/1138 (23.6%) in preadolescence (boys/girls = 135/134) and 869/1138 (76.3%) in adolescence (boys/girls = 498/371), (*p* < 0.05).

Patients were classified according to 25(OH)D into: deficiency [25(OH)D ≤ 20 ng/mL], insufficiency [25(OH)D < 30 ng/mL], and sufficiency [25(OH)D ≥ 30 ng/mL], and based on PTH levels in: PTH ≤ 15 pg/mL (low), PTH ≤ 35 pg/mL (optimal), PTH < 45 pg/mL (normal), PTH ≥ 45 pg/mL (elevated—mild hyperparathyroidism), and PTH ≥ 65 pg/mL (high– overt hyperparathyroidism). Patients were classified by age group into: infancy; <1 year, preschool; 1–5 years, childhood; 6–12 years and adolescence; and 13–16 years. Normal calcium value ranges were considered as 8.5–10.5 mg/dL, Ca > 10.5 mg/dL as hypercalcemia, and < 8.5 mg/dL as hypocalcemia [31]. Τhe results of the study regarding subclinical hyperparathyroidism were compared with pre-pandemic data (2016–2018).

The study was approved by the Institutional Review Board of the Faculty of Medicine of the University of Thessaly, Larisa, Greece (No 6/03-04-2025).

## 3. Results

Vitamin D deficiency was observed in 209/1138 patients (18.3%), while insufficiency occurred in 484/1138 (42.5%). Overall, 60.8% of patients had vitamin D insufficiency and deficiency. Elevated PTH was recorded in 21.5% of patients: 186/1138 (16.3%) with mild hyperparathyroidism and 60/1138 (5.2%) with overt hyperparathyroidism.

Mean 25(OH)D concentrations decreased from infancy through childhood and adolescence (*p* < 0.01) (Table 1). Vitamin D deficiency and insufficiency increased significantly from infancy through childhood and adolescence (*p* < 0.005). Vitamin D deficiency and insufficiency were observed in 23.9% and 40.5% of children aged 13–16 years and 16.9% and in 45.5% of children aged 6–12 years, respectively. At the 1–5 years age group, 36.7% and 16.2% of children were insufficient and deficient, respectively. Regarding infants, 60% had vitamin D sufficiency (Table 1).

In comparison to the pubertal state, the chi-square test revealed a statistically significant difference in vitamin D status and hyperparathyroidism (*p* < 0.001). Specifically, vitamin D sufficiency was 19.9% in adolescents (*p* < 0.001), but it was only 13% in preadolescence (Table 2). Additionally, 18.7% and 6.3% of adolescents, respectively, exhibited mild and overt hyperparathyroidism, compared to 8.5% and 1.8% of preadolescents (*p* < 0.001) (Table 2).

Mean ± SD of 25(OH)D was not statistically significantly different between boys and girls in preadolescence (*p* > 0.05) as opposed to adolescence (*p* < 0.005). (Table 3). Specifically, in adolescence, mean 25(OH)D concentrations were approximately 2 ng/mL higher in males than in females (*p* < 0.05). In addition, PTH was not statistically significantly different between boys and girls in both preadolescence and adolescence (*p* > 0.05) (Table 3).

Of the 186 patients with mild hyperparathyroidism (45 ≤ PTH < 65 pg/mL), a total of 71.4% had abnormal concentrations of vitamin 25(OH)D < 30 ng/mL: 45/186 (24.1%) had vitamin D deficiency and 88/186 (47.3%) insufficiency, while the remaining 53/186 (29.5%) had vitamin D sufficiency. Of the 60 patients with overt hyperparathyroidism, a total of 80% had abnormal concentrations of vitamin 25(OH)D < 30 ng/mL: 25/60 (41.6%) had vitamin D deficiency, 23/60 (38.4%) had insufficiency, while the remaining 20% had vitamin D sufficiency. Of note, from the 61 patients who had low PTH levels ≤ 15 pg/mL: 6/61 (9.8%) had vitamin D deficiency and 24/61 (39.2%) insufficiency. Hypervitaminosis D (25(OH)D > 150 mg/dL [32] was not the underlying cause in any of the 61 patients with hypoparathyroidism. Specifically, from the 29/61 (51%) patients with low PTH, 25(OH)D was 47.7 ± 17.64 with a range [31–107 mg/dL]. More than one-third of patients with hyperparathyroidism had low 25(OH)D concentrations (<30 ng/mL) (Figure 2).

We observed a downward trend of 25(OH)D concentrations in winter and autumn and an increasing trend in summer and spring (*p* < 0.001) (Figure 3).

Regression analysis between 25(OH)D and BMI SDS showed a statistically significant negative correlation (*p* < 0.05)*,* with an impact of 2% (Figure 4).

Regression analysis between PTH and BMI SDS also showed a statistically significant positive correlation (*p* < 0.001) with an impact of 1.5% (Figure 5).

Regression analysis showed a negative correlation between vitamin D and PTH, with PTH starting to rise sharply when vitamin D levels fall below 20 ng/mL and nearly plateauing above 40 ng/mL for 25(OH)D concentrations (*p* < 0.001) (Figure 6).

Subclinical hyperparathyroidism was detected on a significant number of patients: 207/986 (21.5%), 70% of whom had 25(OH)D < 30 ng/mL (Figure 7). Patients with calcium > 10.5 mg/dL were retested to rule out persistent hypercalcemia. All patients had normal calcium levels after retesting.

The data presented above, regarding overall vitamin D status and incidence of sub-clinical hyperparathyroidism, are compared to similar data, previously presented by our group [26], gathered in the pre-COVID era (2016–2018) on 3060 patients, summarized in Table 4. The male-to-female ratio was 40.8% male (64/93) between 2016 and 2018 and 53.3% male (115/131) between 2022 and 2023, with statistically significant more girls than boys (*p* < 0.001). There were also notable differences in vitamin D concentrations between the two periods (Figure 8). Of the patients in the 2016–2018 cohort, 17.8% had deficient levels, 29.0% had insufficient levels, and 52.0% had sufficient levels. Only 29.5% were sufficient by 2022–2023, whereas insufficiency and deficiency rose to 47.3% and 24.1%, respectively (*p* < 0.05 for both insufficiency and deficiency, and *p* < 0.001 for sufficiency). The mean serum 25(OH)D concentrations dropped from 35.20 ± 16.81 ng/mL from 2016–2018 to 26.10 ± 11.21 ng/mL from 2022–2023 (*p* < 0.001) (Figure 8). Mean calcium levels increased slightly from 2022–2023 (9.58 ± 0.55 vs. 9.73 ± 0.75 mg/dL, *p* = 0.05). Although the mean age decreased slightly from 11.56 ± 3.15 to 10.84 ± 3.38 years, the difference was not statistically significant (*p* = 0.06).

## 4. Discussion

Vitamin D is a steroid hormone essential for human health. On the one hand, it contributes to bone health, and on the other hand, as a nuclear steroid hormone in its active form, it has a pivotal role in immunomodulation [33]. Vitamin D insufficiency and deficiency are nowadays considered a pandemic, affecting over 1 billion people worldwide [11]. Differences in vitamin D concentrations vary between countries, which has led to discrepancies in the assessment of vitamin D status in the general population. Yet, the measurement of 25(OH)D concentrations alone without assessing PTH levels provides an incomplete picture of the calcium metabolism itself. Since sun exposure is meant to provide at least 80% of daily vitamin D requirements, the COVID-19 pandemic and the subsequent lifestyle modifications that possibly occurred are likely to have induced a considerable influence on calcium metabolism.

In this study, we retrieved anonymized data on calcium metabolism parameters assessing the occurrence of subclinical hyperparathyroidism—subclinical nutritional rickets (PTH > 45 pg/mL with normal serum calcium) in 1138 children and adolescents in the post-COVID-19 period (2022, 2023) compared to similar data from our group retrieved in the pre-COVID era (2016–2018) on 3060 patients.

Notably, 21.5% of patients had elevated PTH ≥ 45 pg/mL: 186/246 (75%) had mild hyperparathyroidism (PTH ≥ 45 pg/mL) and 60/246 (25%) had overt hyperparathyroidism (PTH ≥ 65 pg/mL). In patients with hyperparathyroidism, a significant 71.4% had 25(OH)D < 30 ng/mL. In mild hyperparathyroidism: 24.2% had vitamin D deficiency and 47.4% insufficiency, and in overt hyperparathyroidism: 41.4% had vitamin D deficiency and 38.4% insufficiency. The optimal PTH is defined as the value where balance is achieved in bone supply and Ca absorption: a value of 15–35 pg/mL indicates optimal bone turnover; a value of 35–45 pg/mL indicates normal bone deposition while maintaining balance, whereas PTH > 45 pg/mL indicates elevated bone resorption [23,30]. Βone turnover markers increase when serum 25(OH)D concentrations are <30 ng/mL, and when serum 25(OH)D concentrations are <20 ng/mL, a fall in serum 25(OH)D causes a rapid increase in PTH [30]. In line with the above, our data show that PTH begins to increase at 25(OH)D concentrations < 40 ng/mL, with the most striking increase observed below 20 ng/mL (Figure 6). Serum PTH has been shown to have an inverse correlation with serum 25(OH)D concentrations in both adults [34] and children in the Mediterranean area [30] and in Northern Europe [30,35].

The adolescent growth period is particularly critical for reaching peak bone mass [36]. To achieve peak bone mass depends on genetic factors, but it is also influenced by dietary, metabolic–endocrine factors, as well as physical activity [36]. Vitamin D and PTH are the main hormonal regulators of bone metabolism and calcium homeostasis [37]. Several factors like vitamin D status, age, BMI, and gender affect PTH levels [38]. The increased frequency of vitamin D deficiency and insufficiency in the general population has an impact on the estimation of the normal PTH range. Studies demonstrate that the highest levels of PTH reduce by 25% after excluding patients with 25(OH)D < 30 ng/mL. These results strongly suggest that vitamin D status should be considered when establishing reference values for serum PTH. Keeping the old reference range with the highest normal level of PTH being 65 ng/mL, cases of mild—subclinical hyperparathyroidism will be missed, underestimating the incidence of the disease [36,38,39].

The diagnosis of Primary Hyperparathyroidism (PHPT) has been based on concomitantly high levels of blood total or ionized calcium (or both) and parathyroid hormone (PTH) [40]. Many patients report with borderline lab results like normocalcemia with high PTH or hypercalcemia with PTH levels that are “inappropriately” within the reference range and yet still have PHPT [41]. The phenotype of normocalcemic or subclinical hyperparathyroidism is now relatively well understood. Despite borderline test results, many individuals experience kidney stones, osteoporosis, and fractures, emerging the need for early diagnosis and treatment. The spectrum of hyperparathyroidism with high serum calcium and yet normal PTH levels is not extensively defined and may involve calcium-sensing receptor variations [41,42].

Subclinical hyperparathyroidism is characterized by elevated PTH levels ≥ 45 ng/mL [30] and normal blood calcium levels in three consecutive measurements over a 3–6 month period after exclusion of causes of secondary hyperparathyroidism [22]. A limited number of studies have been published since 2008, aiming to further characterize this disorder. In two previous studies, Cusano et al. studied the pathogenesis and treatment of the disease [23]. As early as 1975, there was a report on the relationship between vitamin D and PTH, as well as the observation of cases of hyperparathyroidism with normocalcemia, suggesting that vitamin D metabolites may not be mediated solely by changes in serum calcium, and vitamin D metabolites may have a direct effect on parathyroid hormone release [43]. Maruani et al. suggested that the mechanism of persistent normocalciemia, despite high PTH levels, is the resistance to PTH action on bone and kidney [44].

Few population-based studies have examined the incidence of subclinical hyperparathyroidism in healthy population, as PTH values are rarely measured without hypercalcemia or symptoms. Prevalence in a healthy population might be challenging to define. According to accepted diagnostic criteria, the disease’s prevalence ranges from 0.1% to 0.7% [23,45,46]. Few studies have examined the natural history of subclinical hyperparathyroidism, with results from limited cohorts indicating that many patients may not develop hypercalcemia [23,47,48]. A study of 20 patients with subclinical hyperparathyroidism found that none developed hypercalcemia over a 4-year period [48]. In the Dallas Heart Study, only 1 out of 64 patients with subclinical hyperparathyroidism developed hypercalcemia after 8 years. It is claimed that subclinical hyperparathyroidism is a precursor variant of classical PHPT [45]. To the best of our knowledge, this is the first study that investigates the incidence of the disease in children and adolescents. Emerging data suggest that both subclinical hyperparathyroidism and hypoparathyroidism are likely to be asymptomatic in most patients, with a very low risk of progressing to overt disease [45], although the long-term risks of both subclinical hyperparathyroidism and hypoparathyroidism are yet to be defined [22].

The data presented above on the incidence of subclinical hyperparathyroidism were compared with similar pre-COVID-19 data collected by our group between 2016 and 2018, involving 3060 healthy children, summarized in Table 4 [26],. In the 2016–2018 cohort, the incidence of subclinical hyperparathyroidism—elevated PTH with normal calcium—was 5.1%, detected in 154 children. During that period, the male-to-female ratio was 40.8% male (64/93), which increased to 53.3% male (131/115) from 2022–2023 (*p* < 0.001). Notable differences in vitamin D levels were observed between these cohorts: in 2016–2018, 17.8% of patients had deficient vitamin D levels, 29.0% were insufficient, and 52.0% were sufficient, whereas by 2022–2023, sufficiency dropped to 29.5%, with insufficiency and deficiency rising to 47.3% and 24.1%, respectively (*p* < 0.05 for insufficiency and deficiency, *p* < 0.001 for sufficiency). The mean serum 25(OH)D level significantly declined from 35.20 ± 16.81 ng/mL to 26.10 ± 11.21 ng/mL (*p* < 0.001). Mean calcium levels increased slightly (9.58 ± 0.55 vs. 9.73 ± 0.75 mg/dL, *p* = 0.05), while mean age decreased marginally from 11.56 ± 3.15 to 10.84 ± 3.38 years (*p* = 0.06). In the 2016–2018 cohort, almost all cases normalized PTH levels after treatment with cholecalciferol ± calcium at doses used for nutritional rickets. In cases refractory to this treatment after 3 months, adding paricalcitol (0.05 mcg/day) successfully normalized PTH and improved bone health. Notably, even children with vitamin D sufficiency responded to treatment, indicating that normal calcium metabolism is not assured solely by 25(OH)D concentrations ≥ 30 ng/mL. This highlights the necessity of measuring PTH alongside calcium, phosphorus, alkaline phosphatase (ALP), and 25(OH)D for comprehensive bone metabolism assessment [49]. Similarly, evaluating thyroid function requires concurrent measurement of TSH and free T4 (FT4). In complete accordance with our results, Lu et al. strongly suggested taking Ca and vitamin D supplements when serum 25(OH)D is below 20 ng/mL [26]. Finally, according to our results, the prevalence of subclinical hyperparathyroidism in children and adolescents in our country has quadrupled in the post-pandemic period compared to the pre-pandemic era [26].

In our study, 18.3% and 42.5% of patients had vitamin D deficiency and insufficiency, respectively. Notably, over 60% of participants had 25(OH)D values < 30 ng/mL, with no statistically significant difference between boys and girls. A statistically significant negative correlation, though, was found between the incidence of vitamin D insufficiency and age (*p* < 0.05): patients in the age groups 1–5 years, 6–11 years, and 12–16 years had lower 25(OH)D concentrations, with 52.9%, 62.4%, and 64.4% having vitamin D deficiency or insufficiency, respectively. However, during infancy (<1 year), 60% of children had vitamin D sufficiency with a mean 25(OH)D at 38.7 ng/mL, acknowledging the small number of patients in this age group (*n* = 33). A major factor that probably contributes to higher 25(OH)D during infancy compared to childhood and adolescence is the global recommendation for 400 IU/day of vitamin D since birth for all children (irrespective to breastfeeding) [50].

Our findings also demonstrated a statistically significant relationship between vitamin D deficiency, adolescence, and sex. No significant differences in mean 25(OH)D levels were found between boys and girls in preadolescence. However, during adolescence, girls had mean 25(OH)D concentrations approximately 2 ng/mL lower than boys (*p* < 0.05). These results align with previous studies. In a cohort of 384 healthy children (2019–2020) aged 7–16 years, more than 50% had 25(OH)D levels below 30 ng/mL, with a higher prevalence observed among younger children (7–11 years), girls, and those with overweight or obesity [51]. Similarly, a meta-analysis (2000–2022) involving over 7 million individuals across 81 countries reported that 76.6% had 25(OH)D concentrations < 30 ng/mL, with a slight decline in deficiency prevalence from 2011 to 2022, and females being more susceptible than males [52]. Cediel et al. also observed that serum 25(OH)D concentrations decreased from 32.2 ± 8.9 ng/mL before puberty (Tanner I) to 25.2 ± 8.3 ng/mL at puberty onset (Tanner II) [52,53]. Multiple factors contribute to the lower vitamin D levels often observed in girls compared to boys. Sunlight exposure is a primary determinant of vitamin D synthesis, and girls may receive less exposure due to behavioral and sociocultural factors, such as wearing more concealing clothing or spending less time outdoors, especially in regions with conservative dress norms or cultural preferences for lighter skin. Additionally, body composition plays a role, as girls typically have a higher percentage of body fat, which can sequester vitamin D and reduce its bioavailability [8]. Hormonal differences during puberty may also influence vitamin D metabolism, though this area remains under investigation. Dietary patterns further contribute, as girls may be more likely to restrict certain foods rich in vitamin D, such as fortified dairy products, due to body image concerns or nutritional preferences [13]. Hormonal differences, particularly those that emerge during puberty, may contribute to lower vitamin D levels in girls compared to boys, though this relationship is complex and not fully understood. Estrogens—female sex hormones that rise during puberty—can influence the expression of enzymes involved in vitamin D metabolism. For instance, estrogen has been shown to upregulate 1-alpha-hydroxylase, the enzyme that converts 25(OH)D to its active form (1,25(OH)_2_D), potentially increasing vitamin D utilization and altering circulating levels [54,55].

In our study, regression analysis between 25(OH)D and BMI SDS showed a statistically significant negative correlation (*R*^2^ = 0.02, *p* < 0.001) in line with other studies [51,56,57]. Obese individuals may have lower cutaneous production and intestinal absorption, as well as poor metabolism and binding of 25(OH)D in adipose tissue [58], as well as lower sun exposure [59]. Adipose tissue is a primary site for vitamin D storage, and recent studies have shown that VDR and vitamin D-metabolizing enzymes are expressed in adipocytes. Vitamin D influences various functions in adipocytes, including regulating gene expression related to fat cell formation (adipogenesis), and apoptosis in vitro studies indicate that vitamin D stimulates fat production (lipogenesis) and inhibits fat breakdown (lipolysis) via membrane-bound VDR, which is found in caveolae of the plasma membrane. While obesity is associated with vitamin D deficiency, there is no evidence that vitamin D deficiency directly causes obesity. Vitamin D supplementation may help prevent obesity but does not lead to weight loss in obese individuals [60]. Iacopetta et al. have shown an increase in childhood obesity during the pandemic. Sedentary lifestyle, video games, and poor dietary habits are possible explanation of this increase [61].

As anticipated, our data showed seasonal variations, with lower vitamin D concentrations during winter and higher levels during summer, consistent with findings from a Turkish study—a Mediterranean country with similar geographical characteristics—where these seasonal differences persisted despite the overall lower 25(OH)D concentrations during the COVID-19 pandemic [62].

Since sunlight accounts for approximately 80% of daily vitamin D production, the COVID-19 pandemic likely had a significant impact on vitamin D status. According to the World Health Organization (WHO), the pandemic spanned from December 2019 to May 2023. In Greece, where the first confirmed COVID-19 case was reported in February 2020, strict lockdown measures—including movement restrictions, school closures, remote work, and telelearning—were implemented and persisted through 2021. These public health measures substantially limited sun exposure. Despite the well-documented immunomodulatory role of vitamin D and its potential protective effect against SARS-CoV-2 [63], no specific health recommendations for vitamin D supplementation were issued during the pandemic. The pandemic’s profound lifestyle changes—many of which persisted even after the pandemic—are likely to have induced long-term health consequences. Reduced outdoor activities, confinement, and limited access to healthcare and laboratory screening likely contributed to underdiagnosis and undertreatment of hypovitaminosis D. Subsequent studies confirmed an increase in vitamin D deficiency prevalence among children and adolescents following the pandemic [64,65]. For example, a study in India found that 100% of children presenting with symptoms such as myopathy and bone pain had 25(OH)D concentrations < 30 ng/mL after the pandemic [66]. Similarly, research from Brazil reported a decline in adolescents’ mean 25(OH)D concentration from 31.14 ng/mL to 24 ng/mL post-pandemic [67]. In accordance with our results, recent studies suggest that the COVID-19 pandemic has significantly impacted calcium and vitamin D metabolism in children and adolescents, potentially contributing to secondary or subclinical hyperparathyroidism. Reduced sun exposure, dietary changes, and lifestyle shifts during lockdowns have led to widespread vitamin D deficiency, which is a well-known trigger for elevated parathyroid hormone (PTH) levels. For instance, a retrospective study of 348 hospitalized COVID-19 patients found that vitamin D deficiency, particularly when coupled with secondary hyperparathyroidism, was strongly associated with acute hypoxemic respiratory failure and a greater need for ventilatory support [68]. Similarly, a case series reported symptomatic hypocalcemia and high PTH levels in adolescents during the pandemic, directly linked to nutritional vitamin D deficiency [65]. These findings underscore the importance of screening and preventative strategies for vitamin D deficiency and related metabolic imbalances in the post-pandemic era.

Our study has several limitations. Firstly, it was of a retrospective nature, and, more importantly, laboratory testing was not conducted in a single reference laboratory. Additionally, the study did not include measurements of serum magnesium [69] and calcium to creatinine ratio in a morning 2 h urine sample, factors that could influence PTH.

## 5. Conclusions

In conclusion, we found that over 60% of children exhibited vitamin D deficiency or insufficiency, with levels progressively worsening from infancy through adolescence. Alarmingly, more than 21% of children demonstrated subclinical hyperparathyroidism, with over 70% of these cases associated with vitamin D deficiency or insufficiency. Compared to pre-pandemic data (2016–2018), there was a marked deterioration in calcium metabolism parameters, with the incidence of subclinical hyperparathyroidism increasing dramatically from 5.1% to 21.5%. Since subclinical hyperparathyroidism may impair bone mineralization and hinder peak bone mass acquisition, early diagnosis and treatment are crucial. Most cases are secondary and can be resolved with appropriate cholecalciferol and calcium supplementation. Therefore, we recommend concurrent measurement of parathyroid hormone (PTH) and 25(OH)D levels during calcium metabolism assessment in clinical practice. Overall, the COVID-19 pandemic and its lasting effects have significantly worsened vitamin D status and calcium metabolism in children, with potential long-term consequences that likely extend beyond bone health.

## Figures and Tables

**Figure 1 diseases-13-00198-f001:**
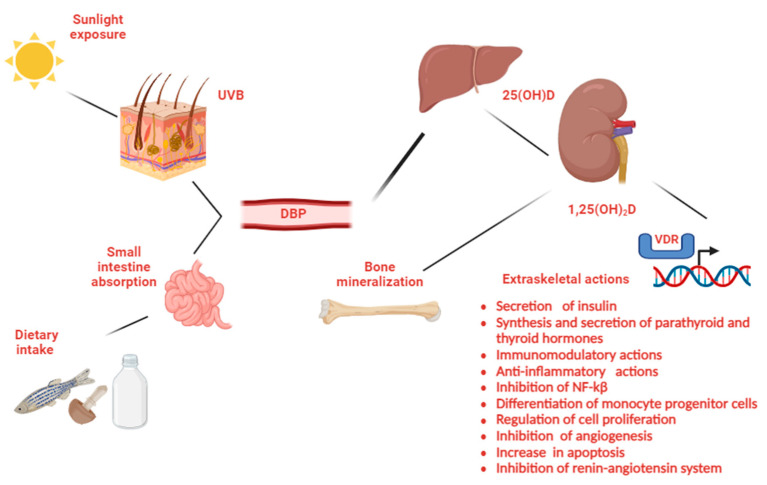
Vitamin D metabolism and actions. (Vitamin D produced in the skin is lipophilic and binds to its transporter protein (VDBP) for transport to the liver, where it is hydroxylated by the enzymes CYP2R1 and CYP27A1 to form 25-hydroxyvitamin D [25(OH)D]. The final activation step occurs in the kidneys, where the enzyme 1α-hydroxylase (CYP27B1) converts the 25(OH)D into the active form, 1,25(OH)2D, calcitriol. Calcitriol is responsible for skeletal and extraskeletal actions of vitamin D) [8,9].

**Figure 2 diseases-13-00198-f002:**
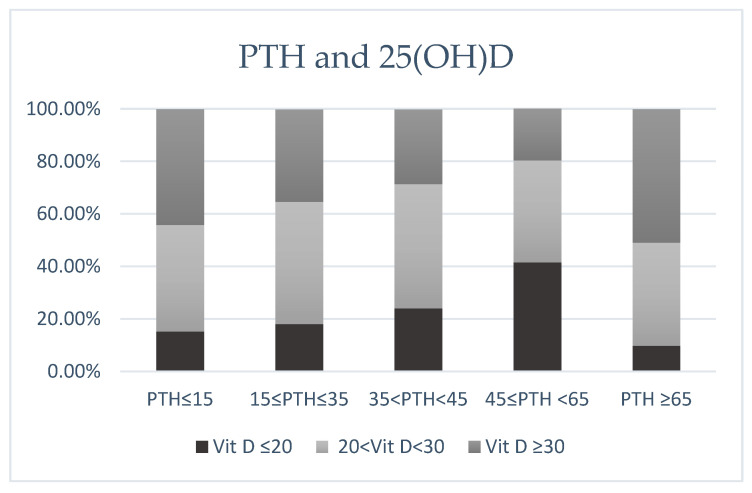
Vitamin D status across PTH (more than one-third of patients with hyperparathyroidism had 25(OH)D < 30 ng/mL).

**Figure 3 diseases-13-00198-f003:**
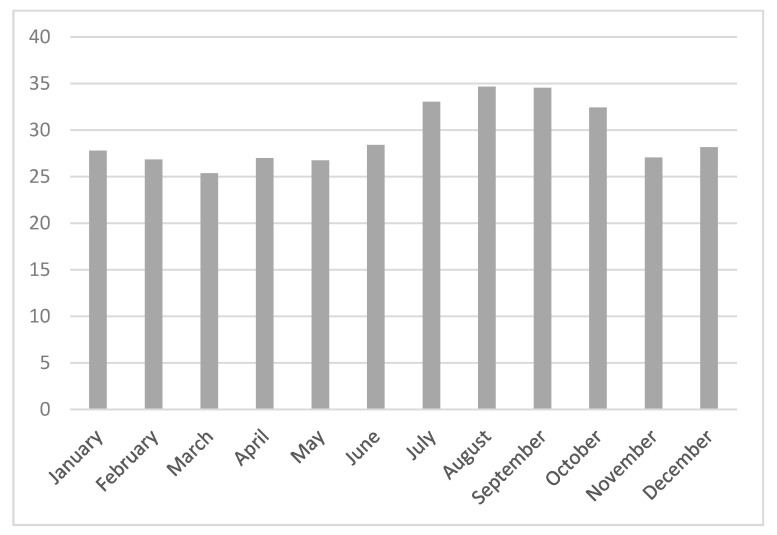
Mean 25(0H)D during months (downward trend of 25(OH)D concentrations in winter and autumn and an increasing trend in summer and spring).

**Figure 4 diseases-13-00198-f004:**
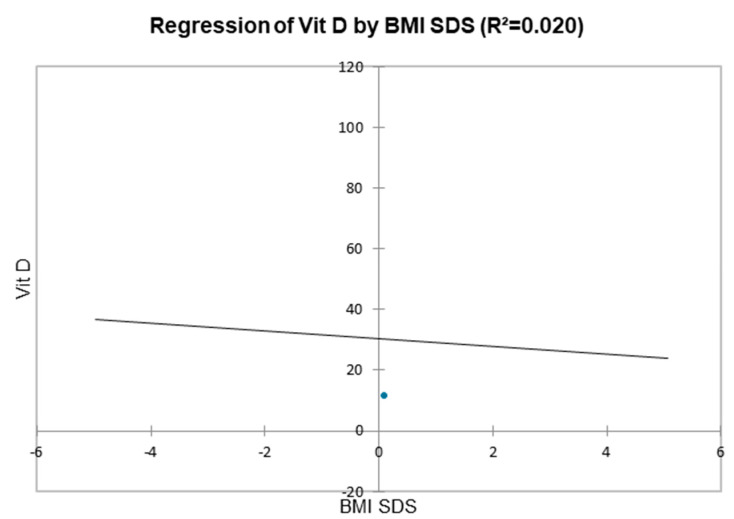
Scatter plot between Vit D and BMI SDS (negative correlation between BMI and vitamin D, meaning that as BMI increases, vitamin D concentrations decrease).

**Figure 5 diseases-13-00198-f005:**
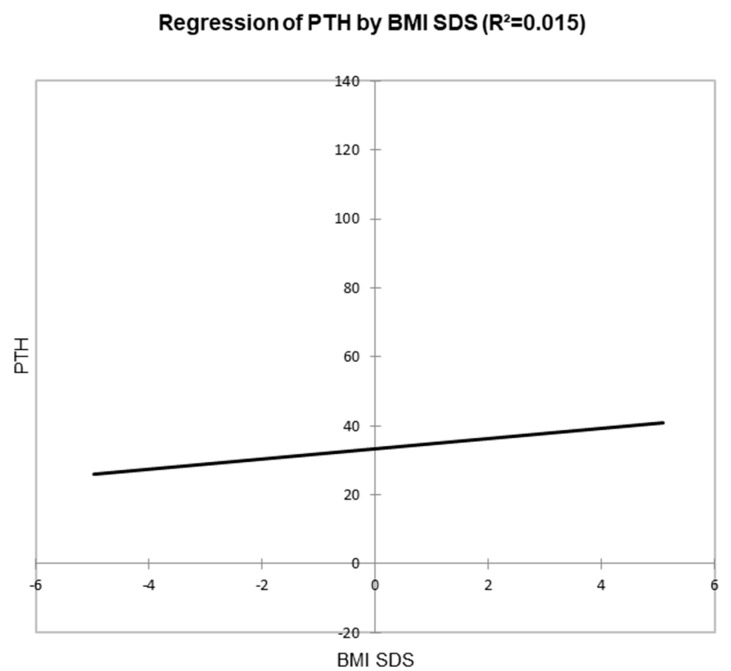
Scatter plot between PTH and BMI SDS (positive correlation between PTH and BMI, meaning that as BMI increases, PTH levels increase).

**Figure 6 diseases-13-00198-f006:**
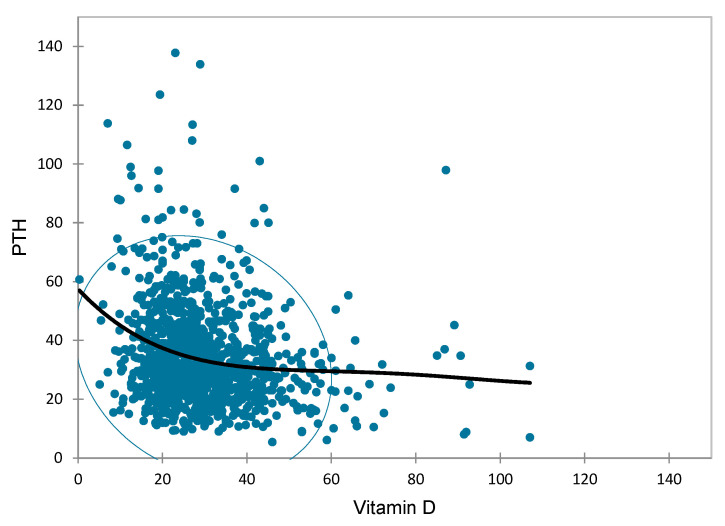
Scatter plot between PTH and Vit D (negative relationship between vitamin D and PTH, with PTH starting to rise sharply when vitamin D levels fall below 20 ng/mL and nearly plateauing above 40 ng/mL for 25(OH)D concentrations).

**Figure 7 diseases-13-00198-f007:**
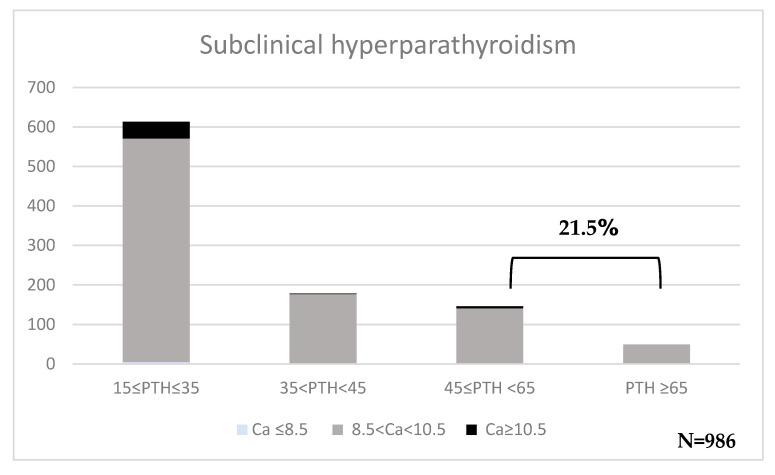
Incidence of subclinical hyperparathyroidism (PTH, pg/mL) regarding calcium levels (Ca, mg/dL).

**Figure 8 diseases-13-00198-f008:**
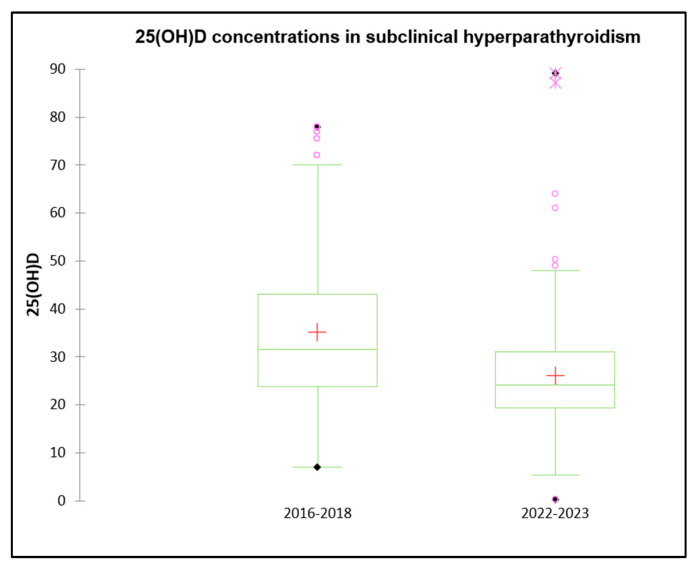
Box plot—25(OH)D concentrations in patients with subclinical hyperparathyroidism (pre and post pandemic).

**Table 1 diseases-13-00198-t001:** Mean 25(OH)D concentrations and incidence of vitamin D deficiency and insufficiency in age groups (Vitamin D deficiency and insufficiency increased significantly from infancy through childhood and adolescence and mean 25(OH)D concentrations decrease, respectively).

Age Groups	*n*	Mean ± SD 25(OH)D	Sufficiency	Insufficiency	Deficiency
<1 year	33	38.70 ± 19.30	60.0%	27.3%	12.1%
1–5 years	166	31.10 ± 13.60	46.9%	36.7%	16.2%
6–12 years	663	29.50 ± 11.50	37.5%	45.5%	16.9%
13–16 years	276	28.70 ± 12.16	35.5%	40.5%	23.9%
*p*-value (between all age groups)		*p* < 0.01	*p* < 0.005		

**Table 2 diseases-13-00198-t002:** Vitamin D status and hyperparathyroidism regarding to pubertal status (*p* < 0.001).

	Preadolescence *n* = 269	Adolescence *n* = 869
**25(OH)D**		
Sufficiency	13.0%	19.9%
Insufficiency	36.8%	44.3%
Deficiency	49.8%	35.7%
*p*-Value	*p* < 0.001	
**PTH**		
PTH ≤ 15 pg/mL	11.8%	3.3%
15 < PTH < 35 pg/mL	63.9%	52.3%
35 ≤ PTH < 45 pg/mL	13.7%	19.2%
45 ≤ PTH < 65 pg/mL	8.5%	18.7%
PTH ≥ 65 pg/mL	1.8%	6.3%
*p*-Value	*p* < 0.001	

**Table 3 diseases-13-00198-t003:** Mean ± SD (range) of 25(OH)D (ng/mL) and PTH (pg/mL) according to sex and pubertal status.

	**Preadolescence** ***n* = 269**	
	**25(OH)D Mean ± SD** **[Range]**	**Mean ± SD PTH** **[Range]**
Girls	33.42 ± 12.68 [9.46–90.60]	28.92 ± 13.68 [6.10–88.10]
Boys	31.26 ± 15.02 [8.34–107.00]	27.20 ± 12.64 [5.40–71.60]
*p*-Value	*p* > 0.05	*p* > 0.05
	**Adolescence** ***n* = 869**	
	**25(OH)D Mean ± SD** **[Range]**	**Mean ± SD PTH** **[Range]**
Girls	27.90 ± 12.34 [0.30–107.00]	37.62 ± 17.74 [9.80–113.80]
Boys	29.52 ± 11.24 [5.90–85.00]	36.20 ± 18.84 [9.30–137.80]
*p*-Value	*p* < 0.05	*p* > 0.05

**Table 4 diseases-13-00198-t004:** Subclinical hyperparathyroidism during the pre-COVID (2016–2018) and post-COVID (2022–2023) periods.

	2016–2018 (*n* = 157)	2022–2023 (*n* = 246)	*p*-Value
Sex (M/F)	64/93 (40.8% M)	115/131 (53.3% M)	<0.001
Age (years)	11.56 ± 3.15 [0.94–16.00]	10.84 ± 3.38 [0.10–16.00]	0.06
25(OH)D (ng/mL)	35.20 ± 16.81 [7.00–78.00]	26.10 ± 11.21 [5.0–89.10]	<0.001
Calcium (mg/dL)	9.58 ± 0.55 [7.90–10.89]	9.73 ± 0.75 [8.10–9.73]	0.05
Vitamin D Status			
Sufficiency	52.0%	29.5%	<0.001
Insufficiency	29.0%	47.3%	<0.05
Deficiency	17.8%	24.1%	<0.05

## Data Availability

The authors will provide the underlying data used in this study upon request.

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
