# Peer review of "Dramatic Deterioration of Subclinical Hyperparathyroidism in Children and Adolescents During the Post-COVID-19 Period"

_diseases, 2025, doi:10.3390/diseases13070198_

Round 1
Reviewer 1 Report
Comments and Suggestions for Authors
While the study addresses a topic of general interest, it does not stand out for its originality. Both the introduction and discussion sections should be revised: the introduction should be more concise and focused, and the discussion should be streamlined to improve its clarity and communicative impact.
The observed differences between male and female adolescents are relevant and quite new. However, these findings should be better contextualized within the discussion. It is recommended that the authors explore possible explanations related to hormonal factors and (where possible), social or behavioral factors (e.g., dietary habits).
The aim of the study needs to be rewritten for greater clarity and coherence. At present, the manuscript states:
….“The aim of the present study was to assess the major parameters of calcium metabolism, such as the incidence of vitamin D deficiency and insufficiency as well as of subclinical hyperparathyroidism in children and adolescents in the post-COVID-19 period (2022 and 2023) compared to relevant respective data from the pre-COVID era (2016–2018)….”
However, in the conclusion, it is stated:
…“This study evaluated the vitamin D status and the incidence of subclinical hyperparathyroidism in the pediatric population in Greece during the post-COVID era.”…
This discrepancy should be addressed and revised to ensure consistency between the stated aims and the conclusions.
Furthermore, the comparisons presented in the tables and the main text are unclear. The authors should clearly define the comparison groups. It is not evident whether:
a) the comparison was made within the same individuals, followed longitudinally from preadolescence during the pandemic to adolescence in the post-pandemic period, or
b) the study involved two independent groupsof subjects of similar age ranges, analyzed in two distinct periods (2016–2018 vs. 2022–2023).
Clarifying this methodological aspect, and aligning it clearly with the study’s aim, is essential for the correct interpretation of the findings.
To support the clarity of the results, the inclusion of a summary figure or graph is strongly recommended. This should visually highlight statistically significant differences in the key biochemical parameters between the two periods (2016–2018 vs. 2022–2023), making the data more accessible and easier to interpret.
Reviewer 2 Report
Comments and Suggestions for Authors
The authors of the manuscript present data from a study that assessed vitamin D status and the incidence of subclinical hyperparathyroidism in the pediatric population in Greece in the post-COVID era. The results show that over 60% of children have a deficiency or insufficiency of vitamin D, with progressive deterioration from infancy to adolescence. The authors found that the frequency of subclinical hyperparathyroidism in children and adolescents in has increased significantly, from 5.1% pre-pandemic to 21.5% post-pandemic, and recommend the simultaneous measurement of PTH and 25(OH)D concentrations. The research topic is current, the methods are relevant. The authors point out the weaknesses of their study. The obtained data can be a basis for future prospective cohort studies.
Reviewer 3 Report
Comments and Suggestions for Authors
Vitamin D is a steroid hormone, essential for the immune system and bone health. Subclinical hyperparathyroidism 21.5%of 1138 children during 2022–2023 were more than that of 5.1% for the pre-pandemic data (2016–2018), which has marked deterioration health in vitamin D status and calcium metabolism in children. These results have important theoretical significance and provide new insights for children and adolescents improve health during the post-COVID-19 period. The analysis process is comprehensive and the article is organized, smooth language and so on. Major revision can be published in Diseases. However, there are some major issues need to be improved:
- Introduction: The preface should supplement the health effects of vitamins and other functional components on COVID-19. For example, please refer to https://www.mdpi.com/1420-3049/29/13/3110
- Materials and Methods: Vitamin D improves COVID-19 literature?
- Results: The title of the figure should be placed below the figure and in a standardized format; Add some subheadings to highlight the highlights and improve readability.
- Discussion: Add some subheadings to highlight the highlights and improve readability. Through the comparative analysis of similar studies at home and abroad, it highlights the highlights and its complex restrictive factors.
- References: Some old references with low correlation can be reduced, but the latest references can be increased.
Round 2
Reviewer 1 Report
Comments and Suggestions for Authors
The additions provided are sufficient to support the publication of the work